# *Legionella pneumophila* Presence in Dental Unit Waterlines: A Cultural and Molecular Investigation in the West Bank, Palestine

**DOI:** 10.3390/tropicalmed8110490

**Published:** 2023-10-30

**Authors:** Ashraf R. Zayed, Mutasem Burghal, Suha Butmeh, Ascel Samba-Louaka, Michael Steinert, Dina M. Bitar

**Affiliations:** 1Department of Biomedical Sciences, Faculty of Medicine and Health Sciences, An-Najah National University, Nablus P.O. Box 7, Palestine; 2Microbiology Research Laboratory, Faculty of Medicine, Al-Quds University, Abu-Dies, East Jerusalem P.O. Box 51000, Palestine; mutasem.burghal@gmail.com (M.B.); salbutmeh@staff.alquds.edu (S.B.); 3Laboratoire Ecologie et Biologie des Interactions, Université de Poitiers, UMR CNRS 7267, 86000 Poitiers, France; ascel.samba@univ-poitiers.fr; 4Institut für Mikrobiologie, Technische Universität Braunschweig, 38106 Braunschweig, Germany; m.steinert@tu-bs.de

**Keywords:** DUWL, *Legionella pneumophila*, West Bank, dental clinics

## Abstract

A *Legionella pneumophila* bacterium is ubiquitous in water distribution systems, including dental unit waterlines (DUWLs). Legionellosis is atypical pneumonia, including Legionnaires’ disease (LD) and the less acute form of Pontiac fever. Legionellosis occurs as a result of inhalation/aspiration of aerosolized *Legionella*-contaminated water by susceptible patients, health workers, and dentists. In this study, we undertook to determine the prevalence of *Legionella* in water and biofilm samples from Tap and DUWLs collected from five sites of dental clinics and faculties across the West Bank. Water samples were tested for physical and chemical parameters. The study samples included 185 samples, 89 (48%) water samples, and 96 (52%) biofilm swabs, which were analyzed by cultivation-dependent analysis (CDA) and by the cultivation-independent technique (CIA). Also, partial sequencing of the 16S rRNA gene for fifteen *L. pneumophila* isolates was performed for quality assurance and identification. *L. pneumophila* was isolated from 28 (15%) of 185 samples using CDA and was detected in 142 (77%) of 185 samples using CIA. The abundance of culturable *L. pneumophila* was low in DUWL of the sampling sites (range: 27–115 CFU/Liter). PCR was 5× more sensitive than the culture technique. *L. pneumophila* Sg 1 was detected in (75%) of the isolates, while (25%) isolates were *L. pneumophila* Sg 2–14. All fifteen sequenced *Legionella* isolates were identified as *L. pneumophila* ≥ 94.5%. The analysis of phylogenetic tree showed that *L. pneumophila* branch clearly identified and distinguished from other branches. These results show that DUWLs of the examined dental clinics and faculties are contaminated with *L. pneumophila*. This finding reveals a serious potential health risk for infection of immunocompromised patients and dentists’ post-exposure.

## 1. Introduction

*Legionella* is a ubiquitous, aquatic, opportunistic Gram-negative bacteria found in many ecosystems, including surface water, groundwater, and water systems made by humans. *Legionella* is responsible for a severe disease called legionellosis. *Legionella* causes nosocomial and community-acquired pneumonia [1]. Legionellosis occurs via the inhalation of aerosolized droplets of *Legionella*. LDis non-communicable and does not spread from person to person [2].

The genus *Legionella* currently includes 59 species. Around half of the species have been found to be pathogenic to humans, and the majority are considered virulent [3]. *L. pneumophila* is responsible for almost 90% of human disease. *L. pneumophila* is classified into 15 serogroups (Sgs) of which Sg 1 is the most prevalent and is responsible for up to 90% of *L. pneumophila* infections [3].

Dental Unit Waterline (DUWL) is part of the environment, which allows *Legionella* and other aquatic bacteria to grow. The problem of bacterial water contamination and biofilm formation in DUWLs has been examined since the moment the first dental chair units (DCUs) were built [4] if not controlled. This was followed by many studies describing both chemical (sodium hypochlorite, chlorhexidine gluconate, hydrogen peroxide) and mechanical (rinsing, filtration) control of microorganisms or biofilm in DUWL [5,6].

*Legionella* typically enters the waterlines of the dental unit (DUWL) from the water supply, where they can proliferate in the biofilm [7,8]. Dentists have a higher prevalence of *L*. *pneumophila* infections compared to other individuals. Patients and dentists are usually exposed to aerosols produced by water spray and handpiece linked to the dental unit. Contaminated water can also access the waterlines of the dental scalers and expose other patients as well as the dentists. Thus, it is of extreme importance to monitor and control possible microbial contamination of this water [8]. Patients and dentists are both exposed to direct contact with bacteria-contaminated water in the form of splatter and contaminated water aerosol sprayed through dental treatment during work by unit handpieces, including rotating and ultrasonic instruments [4,9]. The aerosol droplets generated by dental handpieces are large enough and stable to penetrate the lung alveoli [10]. Key risk factors for developing Legionellosis pneumonia primarily affect individuals with compromised cellular immune systems, chronic heart or lung conditions, renal failure, advanced age, as well as those receiving cortisol treatment, cigarette smokers, and alcohol abusers. [2].

Controlling *L. pneumophila* in DUWLs requires a comprehensive strategy that combines various techniques [7,10]. Firstly, chemical disinfection using substances like chlorine dioxide and iodine tablets is essential [6,11]. Implementing flush protocols, both at the beginning of the day and between patients, helps prevent bacterial buildup [12]. Using distilled or treated water, removing biofilm through enzymatic cleaners, and installing filters are key steps [5,6]. Maintaining proper water temperatures and conducting routine maintenance are crucial. Monitoring plays a vital role, involving water testing, biofilm sampling, temperature monitoring, and meticulous record-keeping [13,14]. Staff training, water quality certification, and a well-defined response plan for positive test results are also essential components. Consulting with experts when needed ensures effective *L. pneumophila* control in DUWLs, prioritizing the safety of dental personnel and patients.

This approach for control is recommended in the national guidelines for most European industrial countries, particularly those who encounter cases of LD [15]. In addition to the health recommendations for dental surgeries, Italian guidelines for the prevention of LD published in 2015 recommended tracking their DUWLs at least annually in order to ensure that the DUWL is free from *Legionellae* [16].

Since there are no previous studies regarding the prevalence of *Legionella* in water and biofilm samples from DUWL in dental clinics in Palestine, our research aims to fill this knowledge gap by conducting a comprehensive investigation into the presence and distribution of this bacterium in dental settings across the West Bank., we carried out this study to evaluate the microbiological quality of Dental Unit Waterline (DUWL) and to determine the prevalence of *Legionella* spp. from DUWL, tap water, and biofilm samples collected from two colleges of dentistry, the clinics in Al-Quds University, Arab American University in Jenin and dental clinics across the West Bank using both standard culture methods and molecular method (PCR) [17]. 

## 2. Materials and Methods

### 2.1. Study Sites, Water and Biofilm Sampling

Water samples and biofilm swabs were collected from various locations, including the dental faculty in Abu Dis/East Jerusalem (31°75′ N, 35°25′ E), the dental faculty in Jenin (32°23′ N, 35°19′ E), as well as dentists’ clinics in different regions within the West Bank (WB), such as Nablus (32°22′ N, 32°13′ E), Tulkarem (32°31′ N, 32°18′ E), and Hebron (31°52′ N, 31°31′ E). (Appendix A). Water and biofilm samples were analyzed for the presence of any bacteria, *Legionella* spp. and *L. pneumophila* in their water sources. Water samples were collected, processed, and analyzed according to international standard operational procedures (SOPs). For water quality, detection, and enumeration of *Legionella*, ISO 11731 were used [18]. For water quality, detection, and enumeration of *Legionella* part two (direct membrane filtration method for waters with low bacterial counts), ISO 11731-2 was used [18]. 

From February through October 2018, a total of 89 water samples were collected. One liter of each DUWL and tap water was collected for DNA extraction. Also, one liter of each DUWL and tap water was collected for *Legionella* count. Tap water samples were collected in sterile 1 L polyethylene bottles after a brief flow time (2–3 min) to permit clearing the service line. Water flow was reduced to permit filling the bottles without splashing. To neutralize residual free chlorine, 0.5 mL of 0.1 M sodium thiosulphate (Na_2_S_2_O_3_·5H_2_O) (0.1 g/100 mL) was added to the sterile bottles for *Legionella* count analysis [19].

From the DUWL, sampling was carried out by collecting and mixing about 200 mL of water (for a total of 1 L) from each of the following: 1. Air-water syringe, 2. Micro-engine, 3. Turbine, 4. Ablator, 5. Cup filler.

The water samples were kept at a refrigeration temperature of 4 to 8 °C until analysis, culture, and DNA extraction. Samples were delivered to the Microbiology Research laboratory within one day. Temperature, pH, and conductivity were measured on-site for every water sample.

Also, a total of 96 biofilm swabs from anterior surfaces of faucets of DUWL and clinic taps were obtained for DNA extraction using sterile cotton swabs (Cotton Tipped Applicator, Beijing, China) and for *Legionella* identification using transport medium (Copan, Culture swab transport system, Italy). Samples were taken from each site randomly. Biofilm swabs for *Legionella* identification were processed in the laboratory by culturing on BCYE (CM0655, OXOID, UK) and GVPC (Glycine-Vancomycin-Polymyxin-Cycloheximide) (CM0655, OXOID, UK) medium immediately. The swabs for DNA extraction were kept at −80 °C until DNA extraction was performed.

### 2.2. Measurement of Physical and Chemical Background Parameters

Tap and DUWL water samples were tested for temperature using an electronic thermometer (ama-digit, ad 15th, Hannover, Germany), pH measurement, and conductivity using PCE meter (PCE-PHD 1, Hannover, Germany) on site. Upon the arrival at the Microbiology Research Laboratory, water samples were tested for total chlorine (Cl_2_) and total hardness in water (content of calcium and magnesium salts) using quantofix sticks according to the manufacturer’s instruction (Quantofix, Macherey-Nagel GmbH & co.KG, Dueren, Germany). Total chlorine is the sum of all free and combined chlorine (Chloramines) species. Quantofix strips is a semi-quantitative method to perform chlorine measurements. These test strips combine fast performance with accurate results. They are based on a chemical reaction between chlorine and dye. A redox reaction takes place, and the test pads develop a certain color intensity depending on the concentration of chlorine in the sample (Quantofix, Macherey-Nagel GmbH & Co. KG, Dueren, Germany).

### 2.3. Cultivation Dependent Analysis (CDA) of Water and Biofilm

Each water specimen was filtered onto a membrane filter (pore size 0.22 µm, diameter 47 mm, MILLIPORE, Ireland) using a sterile filtration unit (Nalgene, Schwerte, Germany). The vacuum pump (LVP 500, Sejong-Si, Republic of Korea) pressure was approximately 200 mbar. For acid treatment, 30 mL of acid buffer was added on top of the membrane filter and was left for 5 min. The filter was then rinsed with 20 mL Page’s saline. Page’s saline is recommended for bacterial concentration including *Legionella* organisms by membrane filtration for *Legionella* detection and enumeration. The membrane was then aseptically placed onto the agar plate. Triplicates of BCYE and/or GVPC (CM0655, OXOID, Hampshire, UK) agar plates were used with a chemical enrichment supplement recommended for enhancing *Legionella* species growth. The plates were used according to the manufacturer’s instructions. The plates were incubated inverted at (36 ± 2) °C for 10 days. Plates were checked for growth twice, on the third or fourth day for ten days. The final reading was performed after ten days with a description of the colonies. More details on cultivation-dependent analysis are given by Burghal MZH and Zayed et al. [17,20]. 

### 2.4. Cultivation Independent Analysis (CIA) of Water and Biofilm (16S rDNA PCR)

A total of 89 samples of water were collected for DNA extraction. Water samples were filtered onto sandwich membrane filters composed of nucleopore filter (Nuclepore Track-Etch Membrane, MB 90 mm, 0.2 µm, Whatman, Buckinghamshire, UK) and glass fiber-microfilter (GF/F) (GFF, 90 mm, Whatman, Buckinghamshire, UK). Also, a total of 96 biofilm swabs from the anterior surfaces of faucets of DUWL and clinic taps were obtained for DNA extraction using sterile cotton swabs (Cotton Tipped Applicator, Beijing, China). For the extraction of DNA from the filter sandwiches and the swabs, a modified DNeasy protocol (Qiagen kit No. 69506, Hilden, Germany) was used. Briefly, sandwich filters were cut into small pieces and incubated with enzymatic lysis buffer (20 mM Tris-HCl, 2 mM EDTA, 1.2% Triton X-100 (pH 8.0)) containing 10 mg/mL lysozyme for 60 min in a 37 °C water bath. After the addition of AL buffer from the kit, the samples were incubated at 78 °C in a shaking water bath for 20 min. After filtration through a cell strainer, i.e., 100 µm (DB falcon 352,360, Corning, Glendale, AZ, USA), absolute ethanol was added to the filtrate (ratio of filtrate to ethanol is 2:1), and the mixture was applied to the spin column of the kit. After this step, the protocol was followed according to the manufacturer’s instructions. Three different PCRs were carried out as follows: (i) for the detection of any bacteria, the bacterial common 16S rRNA gene primers (Com), (ii) for *Legionella* genus-specific primers (Lgsp), and (iii) for *L. pneumophila* species-specific primers (Lp1) were applied [21]. Each PCR reaction was carried out using 3 µL (1 ng/µL) of DNA template in a final volume of 25 µL. Amplification was achieved using PCR-ready Master Mix (GoTaq, Green Master Mix, Promega, Madison, WI, USA). To test the specificity of *L. pneumophila* primers and confirm species identity, fifteen isolates were identified by amplifying and sequencing an internal fragment of the 16S rRNA gene, according to Senderovich et al. [22]. The obtained sequences were compared using the NCBI service to certain closest relatives. The sequences were submitted to the GeneBank BioProject database (PRJNA1002630) and BioSample (SAMN36886996-SAMN36887010). Sequencing of the 16S rRNA gene of the six isolates confirmed the presence of *L. pneumophila* (≥94.5% 16S rRNA gene similarities).

### 2.5. Serogrouping of Legionella Isolates

The serogroups of the 28 *L. pneumophila* isolates were identified by an agglutination test using *Legionella* Latex (Oxoid DR0800, Basingstoke, UK). Using this test, the isolates were sero-grouped as Sg 1 and Sg 2–14. The Oxoid *Legionella* Latex Test is a latex agglutination test for the identification of predominant *Legionella* species grown on plate media from patients with suspected Legionellosis or from environmental sources. The Oxoid Legionella Latex Test allows separate identification of *L. pneumophila* Sg 1 and Sg 2–14 and detection of seven other *Legionella* species (Legionella *longbeachae* 1 and 2 *Legionella bozemanii* 1 and 2 *Legionella dumoffii Legionella gormanii Legionella jordanis Legionella micdadei Legionella anisa*) which have been implicated in human disease (Oxoid DR0800, Basingstoke, UK).

### 2.6. L. pneumophila 16S rRNA Sequencing and Phylogenetic Analysis

To test the specificity of *L. pneumophila* primers and confirm species identity, fifteen isolates were identified by amplifying and sequencing an internal fragment of the 16S rRNA gene according, to Senderovich et al. [22]. The obtained sequences were compared using the NCBI service to certain closest relatives. The sequences were submitted to the GeneBank database (SAMN36886996-SAMN36887010) and BioProject accession number PRJNA1002630. Sequencing of the 16S rRNA gene of the fifteen isolates confirmed the presence of *L. pneumophila* (≥94.54% 16S rRNA gene similarities). A detailed list of all isolate accession numbers is provided in Appendix A. Then, the phylogenetic tree analysis was imported into the MEGA11.0.13 software by using the Maximum Likelihood (ML) method with 1000 bootstrapped replicates.

### 2.7. Statistical Analysis

Excel (Microsoft Office, 2019) was used for statistical analysis. Data are presented as means ± standard deviation (SD) and prevalence. BLASTn was used to analyze the results of sequencing. Sequences retrieved from isolates were deposited in the GenBank. Then, the phylogenetic tree analysis was imported into the MEGA11 software [23] by using the Maximum Likelihood (ML) method with 1000 bootstrapped replicates. Any *Legionella* counts (<5 CFU/L) were reported as BD and excluded from statistical analysis (average of *Legionella* count and occurrence frequencies).

## 3. Results

### 3.1. Characteristics of Physico-Chemical Properties and Legionella Count in Tap Water and Drinking Unit Water Line

The sampled drinking water of the five sites was mainly tap water and DUWL and characterized by a high hardness (on average 234–291 mg/L CaCO_3_ equivalents) and high conductivity (on average 724–862 µS). The average temperature of both tap water and DUWL ranged between 18 ◦C and 24 °C. The average pH of the water was 7.2 and 7.9. Chlorine was, on average, 0.5 mg/L (Table 1).

During the study period, *Legionella* counts were detected in the tap water and DUWL of all sampling sites except for Hebron clinics and DUWL in Nablus clinics. Sampling comprised water and biofilms, with comparable numbers of samples taken from each sampling site. From water samples, thirteen *L. pneumophila* strains were isolated from 89 samples (Table 2). *L. pneumophila* was isolated from tap water and DUWL of all sampling sites except for Hebron. For the collection period, the *Legionella* counts per site ranged from 0 to 115 CFU/L (Table 1). In Jenin dental faculty, on average, 57 ± 12 CFU/L of *Legionella* spp. from tap water were detected, and 53 ± 6 of *Legionella* spp. from DUWL were detected; in Tulkarem dental clinics, 115 ± 11 CFU/L of *Legionella* spp. were detected from tap water and 42 ± 5 CFU/L from DUWL; in Nablus dental clinics, 27 ± 9 CFU/L of *Legionella* spp. were detected from tap water only; finally, in Abu Deis/East Jerusalem dental faculty, on average, 47 ± 5 CFU/L *Legionella* spp. were detected from tap water and 54 ± 12 CFU/L of *Legionella* spp. were detected from DUWL (Table 1). *Legionella* spp., mainly *L. pneumophila*, was isolated from 15 out of 96 biofilm swab samples (15.6%). The highest frequency of *L. pneumophila* isolates was detected in Tulkarem dental clinics (22.6%), where *Legionella* spp. were also detected in both tap and DUWL water during the study period. Meanwhile, the lowest frequency was detected in Nablus dental clinics (8.7%) (Table 2). Also, a statistically significant difference (Independent *t*-test *p ≤* 0.01) was shown between tap water and DUWL. For all of the clinics with detections, the tap water samples have higher or comparable *Legionella* counts compared with the DUWLs. This indicates the entire dental clinic water systems are likely contaminated.

### 3.2. Comparing Cultivation-Dependent Analysis (CDA) with Cultivation-Independent Analysis (CIA)

Although CDA is the standard and recommended technique for environmental surveillance of *L. pneumophila*, CIA provides higher sensitivity and overcomes the problems of CDA for *Legionella* because of the VBNC state and its overgrowth by competing bacteria. In this study, both methods were used to detect *Legionella* spp. in dental clinics and faculties. A total of 89 water samples and 96 biofilm swabs from the five sampling sites were tested by conventional PCR using three different primers (com, Lgsp, Lpn). Almost all of the samples were positive using com primers (n = 89, 100% and n = 96, 100%) for water samples and biofilm swabs, respectively. *Legionella* spp. were detected in biofilm swabs more than in water samples (n = 85, 88.5% and n = 62, 69.7%), respectively (Table 2). Similar results were obtained using *L. pneumophila*-specific primers: 86.5% of the biofilm swabs and 66.3% of the water samples were positive. As expected, the PCR-based CIA showed higher sensitivity than CDA. CIA analysis increased the detection of *L. pneumophila* from 14.6% (CDA) to 66.3% (CIA) for water samples and from 15.6% (CDA) to 86.5% (CIA) for biofilm samples. Moreover, *L. pneumophila* isolates were detected in tap water and biofilm water more than in DU water and dental biofilm samples (17.9%, 20% and 12%, 11.8%), respectively, using cultivation-dependent analysis. Similar results were obtained using cultivation-independent analysis (*L. pneumophila*-specific primers): 82.1% and 91.1% of tap water and tap biofilm swabs and 54.1% and 82.4% of dental water and dental biofilm samples, respectively (Table 2).

### 3.3. Distribution of L. pneumophila According to Serogroups in Dental Sites

The twenty-eight PCR-confirmed *L. pneumophila* environmental isolates were tested for serogroups. Most of the isolates were characterized as Sg 1 (Sg 1) (n = 21); the remaining seven isolates were (Sg 2–14) as determined using an agglutination kit (Table 3).

As shown in Table 3**,** out of 11 isolates from Jenin dental faculty, nine belonged to Sg 1, and two belonged to Sg 2–14. Out of seven isolates from Tulkarem, six belonged to Sg 1, and one belonged to Sg 2–14. Also, out of four isolates from Nablus, three belonged to Sg 1, and one belonged to Sg 2–14. Out of six isolates from Abu Deis/East Jerusalem dental faculty, three isolates belonged to Sg 1, and three belonged to Sg 2–14. No isolate was obtained from Hebron by CDA.

### 3.4. 16S RNA Sequencing of L. pneumophila Isolates

Fifteen isolates were subjected to sequencing, revealing their taxonomic affiliation as *L. pneumophila*. We conducted 16S rRNA sequencing to explore the relatedness between our collection of *L. pneumophila* strains and sequences available from the NCBI database. Using the NCBI BLASTn tool, we identified sequence products listed in Appendix A as highly similar to *L. pneumophila* sequences, with an identity value of ≥94.4%. This conclusion was further supported by phylogenetic tree analysis (Figure 1).

Specifically, *L. pneumophila* isolates from Tulkarem DUWL and dental unit biofilm (DUB) (T2_DUWL_Ps and T4_DUB_Ps) clustered together. Similarly, sequences from Tulkarem tap water and biofilm (T1_TW_Ps and T3_TB_Ps), as well as Jenin tap water and biofilm (G1_TW_Ps and G3_TB_Ps), formed distinct clusters. Also, the *L. pneumophila* isolates from Nablus dental clinics (N1_TW_Ps and N2_TB_Ps) appeared to be closely related to each other. Conversely, the *L. pneumophila* isolates from Abu-Deis/East Jerusalem dental faculty pairs (AQU1_TW_Ps, AQU3_TB_Ps, AQU2_DUWL_Ps, and AQU4_DUB_Ps) did not cluster together. The importance of the findings is reflected in the following: (i) this information can provide insights into the diversity of *L. pneumophila* in the West Bank, aiding in epidemiological studies, source of contamination, and outbreak investigations. (ii) The clustering of *L. pneumophila* isolates from specific geographic locations suggests potential regional variations or sources of contamination. Understanding these patterns can inform public health efforts to control and prevent the spread of LD. (iii) It highlights the importance of water quality and biofilm management in preventing potential infections associated with *L. pneumophila* in dental facilities.

Based on the analysis of a phylogenetic tree, we examined thirty-five strains of *L. pneumophila* collected from various geographical locations worldwide. Additionally, we included eleven strains from the following species: *L. dumoffii*, *L. anisa*, *L. wadsworthii*, *L. saoudiensis*, *L. longbeachae*, *L. sainthelensis*, *L. feeleii*, *L. micdadei*, *L. jordanis*, *L. israelensis*, *L. oakridgenesis*, and *L. hackeliae*. This phylogenetic tree was constructed based on the 16S rRNA gene. To further assess our isolates, we obtained isolates from different countries from NCBI and compared them to our samples. The ML phylogenetic tree clearly distinguished the branch containing *L. pneumophila* from the other branches. *L. pneumophila* formed a distinct branch in the tree, enabling us to design species-specific primers and probes for future research and monitoring (Figure 1). This finding supports the ability to design species-specific primers and probes based on the ML phylogenetic tree, which is significant. These primers and probes can be used in diagnostic tests to accurately identify *L. pneumophila* strains. This is important in clinical settings for diagnosing LD.

## 4. Discussion

### 4.1. Legionella Abundance in Dental Water and Biofilm Samples from the West Bank

This study is based on the first sampling study examining the prevalence of *Legionella* spp. in dental clinics and faculties in the West Bank. The analysis of water and biofilm samples was performed using cultivation-dependent and -independent methods targeting *Legionella* from the genus to the species level for *L. pneumophila* by molecular techniques, including conventional PCR and 16S rRNA sequencing [17].

Generally, water samples had a far lower prevalence of *Legionella* compared to biofilms. Water samples tested positive for the presence of *Legionella*, with a prevalence of 14.6% by CDA and 69.7% by CIA. Biofilms had a higher prevalence, with 15.6% positive by CDA and 88.5% by CIA (Table 2). The findings of increased PCR-based detection in water and biofilms are consistent with our previous extensive study from eight hospitals across the West Bank [20]. The proposed sampling strategy helps identify contamination sources and improve maintenance for DUWLs. A multidisciplinary approach is crucial for proper DUWL management, ensuring safety for both personnel and patients while maintaining functionality [9].

All the samples in our study were collected at the beginning of the day to detect the highest level of *Legionella* by CDA. *L. pneumophila* was isolated from DUWL and tap water, as well as from biofilm swabs. The *Legionella* count from the water samples varied between 27 ± 9 CFU/L and 115 ± 11 CFU/L. The American Dental Association (ADA), in 1996, set a limit for DUWL to contain less than 200 CFU/mL. The Center for Disease Control and Prevention (CDC) in 2003 recommended ≤500 CFU/mL for non-surgical dental procedures [12]. Theoretically, the potential health hazard of *Legionella* to humans is associated with cell concentrations above 10^4^ to 10^5^ CFU/L of water [24]. Persistence of *L. pneumophila* in aquatic systems is a health hazard, and this is reflected in the medical research focus on this fastidious bacteria [25]. Recently, an extensive systematic review of bacterial biofilm in DUWL showed that the prevalence of *L. pneumophila* in DUWLs was estimated to be 12% (95%CI: 10–14%) [14]. According to the previous study, our results were in the cut-off value (11.8% in Biofilm dental unit) (Table 2). 

The prevalence of *Legionella* in culture-based studies is in accordance with studies in the West Bank, Italy, and Greece [5,20,26,27,28]. However, many studies showed a much higher culturable *L. pneumophila* prevalence in water, e.g., 21.6%, 22%, and 40% in Kuwait, Tunisia [24,29], and Jordan [30], respectively. The prevalence of *L. pneumophila* was even higher (68.5%) in a study from northern Israel [31]. Almost half of the *L. pneumophila* isolates from the West Bank (n = 15, 60%) were obtained from biofilm samples (Table 2). 

Biofilms provide *Legionella* with nutrients for growth and protection from adverse ecological conditions, such as water disinfection. Ma’ayeh et al. [30] studied the contamination of DUWL water systems with *Legionella* at the University of Science and Technology in Jordan. The rate of detection of *Legionella* from DUWL was 86.7% at the beginning of the day, 40% after 2 min of flushing, 53.5% at midday. Another study reported the rate of detection of *Legionella* Sg 1 from DUWL was 36.1% (9/52): 17.3% at the beginning of the day, 5.7% after 2 min of flushing, 5.7% at midday [8]. Globally, various studies have revealed that DUWLs have high levels of microbial contamination. In 1995, Challacombe and Fernandes studied 194 DUWL to detect the presence of *Legionella*, they found very low concentrations in 49/194 (25%) and 145/194 (75%) were negatives [32]. A study in the USA examined 28 dental clinics in six U.S. states that tested positive for the presence of *L. pneumophila* and other *Legionella* spp. by PCR [33]. Williams et al. studied 47 DUWL biofilms and found 62% of them had *Legionella* and 19% concentration exceeding 100 CFU/ml [34]. A study in Torino/Italy by Ditommaso et al. in 2016 determined the prevalence of *Legionella* in DUWLs and tap water samples by using PMA-qPCR propidium monoazide (PMA) quantitative PCR (qPCR) and standard culture methods. The results showed the level of *Legionella* spp. was very low. Detection of contaminated water by CDA does not reflect the true scale of the problem, so they needed to do a heterotrophic plate count on yeast extract agar, based on the assumption that *Legionella* is part of the components of biofilms and *Legionella* is a fastidious bacterium, affected by overgrowth of other bacteria [16]. Swabbing as a method of biofilm sampling has its drawbacks. Scraping the surface would be a better option and even higher results would be obtained.

CDA for *Legionella* isolates have several limitations that can result in low sensitivity and the failure to detect these bacteria effectively. Here are some reasons why CDA may not be optimal for isolating *Legionella*: (i) *Legionella* bacteria have specific growth requirements. They thrive in aquatic environments and are often associated with biofilms and amoebae. These requirements can make it challenging to cultivate *Legionella* in vitro. (ii) *Legionella* is a slow-growing organism, and it can take several days for colonies to appear on culture plates. This extended incubation period increases the risk of contamination. (iii) *Legionella* can enter a VBNC state under certain conditions, which makes them undetectable by standard culture methods. In this state, they may still be alive and potentially pathogenic but cannot be cultured using conventional techniques. (iv) *Legionella* may not be present in high concentrations in environmental samples. This low prevalence can make it difficult to detect the bacteria using CDA, especially if the sample is not appropriately concentrated or if the *Legionella* cells are in a non-culturable state. Due to these limitations, CIA has become increasingly important for the detection and identification of *Legionella*. CIA can provide faster and more sensitive results, enabling better surveillance and management of *Legionella*-related diseases.

### 4.2. Importance of the Study Findings in Dental Water Management Strategy and Health Impact on Chronic Diseases

DUWL water is an ideal environment for the presence of biofilm and microbial contamination due to the nature of the tubing of the DUWL. This phenomenon has been well documented, as well as the difficulties in biofilm cleaning and the prevention of regrowth [7,10,14]. With regard to exposure of the dental patient, the health worker, and the dentist in a dental clinic, the nature of the use of the DUWL in dentistry helps in the production of aerosols and splatter generated by working handpieces, two important means of transmission of *Legionella* bacteria. The dental patient is also exposed to contamination from DUWL from backward contamination, which may occur when oral normal flora of patients enters the waterlines via suctioning of saliva by the head of the handpiece [10]. Our study showed that tap water (the supply of DUWLs) was generally contaminated for the sample sites. Such contamination in tap water can easily transfer to any water supply system or pipeline, causing dramatical health problems. 

Dentists and patients can be exposed to opportunistic or pathogenic microorganisms, including *Legionella* bacteria, by inhaling droplets and aerosols produced by dental instruments connected to DUWLs [10,14], but the extent of the problem is generally unrecognized, and there are no specific guidelines for protecting patients and dentists from exposure to aerosols contaminated with *Legionella*. Biofilm is a complex heterogeneous microbial cluster that forms on any non-sterile moist surface. Being an aquatic organism, *Legionella* is found growing in the biofilm that lines the inside of pipes and water lines [14,35]. *L. pneumophila* is considered an important pioneer colonizer in aquatic environments, especially DUWL [14]. 

Unfortunately, there are no previous data about legionellosis cases in Palestine with regard to dental patients. Regarding clinical analysis from the West Bank, Jaber et al. showed that there was a high risk of lung infection due to *L. pneumophila* as indicated by the high percentage of infected pneumonia patients [36]. Also, a recent case study of *L. pneumophila* ST461 and Sg 6 caused severe nosocomial LD in a woman with chronic hypertension [37]. Furthermore, we studied the infectivity and cytotoxicity of all *L. pneumophila* clonal complexes and their affiliated genotypes [38]. The globally distributed ST1 showed high virulence characteristics compared to the endemic ST461. However, virulence traits and the overall infection processes are rather complex phenomena. In addition, there are other aspects contributing to the risk of LD, such as the infective dose, ecology, and ecotype characteristics [20,26]. A case report in 2012 talked about a healthy 82-year-old Italian woman who contracted LD after a dental appointment and a report of a fatal case of legionellosis-based pneumonia in a dentist in the United States [16,33]. Mizrahi et al. studied 133 clinical sputum samples from Israel which were positive for *Legionella* by PCR 9/133 (6.8%), and only one sample out of the nine was also positive by culture and belonged to *L. pneumophila* Sg 1 [2].

*L. pneumophila* is the most pathogenic of *Legionella* spp., causing up to 90% of the cases of legionellosis [25,39]. *L. pneumophila* Sg 1 represented 75% of the total isolates, while 25% of the isolates belonged to *L. pneumophila* Sg 2–14 (Table 3). This is in accordance with our previous study, where the most prevalent serogroup was Sg 1 (61.6%) followed by Sg 6 (30%), and the rest of the isolates belonged to other serogroups [20]. According to the current epidemiological data available from the world, different *L. pneumophila* Sgs cause legionellosis. Mavridou et al. studied the prevalence of *Legionella* spp. in Greek hospitals. They found 72.7% of *Legionella* was *L. pneumophila* Sg 1 and 22.7% were *L. pneumophila* Sg 2–14 [28]. Furthermore, a Korean study investigated the distribution of *Legionella* spp. from environmental water sources of public facilities in South Korea. They isolated 560 *Legionella* isolates from all of South Korea. They found 85.5% of the isolates were *L. pneumophila* Sg (54.7%). The rest of the isolates (14.5%) were non-*L. pneumophila* [40], whereas in the Middle East, there is a shortage of epidemiological data for *Legionella* Sgs. A recent study in Israel indicated that *L. pneumophila* Sg 3 might be the primary causative agent responsible for legionellosis [41]. Blanky et al. in Israel revealed 23 water samples were *Legionella*-positive: *L. pneumophila* Sg 1 (87%), serogroup 3 (21%), and serogroups (2, 4–14) (18%) [42]. Similarly, another study in Kuwait on clinical isolates demonstrated dominance (more than 80%) of *L. pneumophila* Sg 3 in patients with LD (Qasem et al., 2008). The second Study in Kuwait by Al Matawah revealed the 46 *L. pneumophila* isolates, the majority of the isolates belonged to serogroup 3 (80%), followed by serogroup 1 (13%), serogroup 7 (2%), serogroup 10 (2%), and serogroup 4 (2%) [24]. 

Tesauro et al. mentioned the health risks of LD in patients with chronic diseases. The author isolated *Legionella* spp. from 12.3% dental plaque samples of two positive patients who have reported Chronic Obstructive Pulmonary Disease (COPD) currently and pneumonia in the past [43]. Recently, Lehfeld et al. warned that wearing dentures or poor oral hygiene might confer an increased risk of acquiring the infection at home, and oral hygiene may prevent acquiring the infection at home [44].

To protect patients and control *Legionella* in healthcare facilities, a comprehensive water management plan is crucial [9,12]. This plan should identify potential sources of *Legionella* growth and transmission within the facility’s water systems, such as cooling towers, air conditioning systems, water storage tanks, hot water tanks, and showers [45,46]. Maintaining proper water temperatures, with hot water stored at 60 °C and cold water below 25 °C, is essential [47]. Regular cleaning and disinfection of water systems, along with the use of appropriate disinfectants, should be implemented [7]. Chlorination or other approved water treatment methods may be necessary in high-risk areas [7]. Regular flushing of stagnant water in low-use areas and the installation of filters to capture *Legionella* are recommended [12]. Routine water sampling and testing using a certified laboratory should be conducted, with prompt responses to positive results [9]. Detailed documentation of water management activities is essential, and staff should be trained in *Legionella* prevention. Effective communication protocols, compliance with regulations, and regular audits and reviews of the water management plan are also crucial [9]. Overall, a multidisciplinary approach involving facility management, infection control teams, water treatment experts, and public health authorities is necessary to protect patients and staff from LD. This study illustrates the importance of protecting patients, dental health workers, and dentists from contamination with *L. pneumophila* bacteria by inhalation of aerosols, which may cause LD, and highlights the need for appropriate specific guidelines for protecting patients from exposure.

### 4.3. Comparing L. pneumophila 16S rRNA Sequences from the West Bank to the International Database

To address the distribution of the fifteen *L. pneumophila* isolates of the West Bank, the 16S rRNA sequences were compared to the *L. pneumophila* 16S rRNA NCBI database available from different countries (https://www.ncbi.nlm.nih.gov/genome/browse#!/prokaryotes/416/, accessed on 26 August 2023) (Figure 1 and Appendix A). Our recent study described a set of thirty-eight clinical and environmental *L. pneumophila* genomes retrieved from Germany and the West Bank [48]. Also, the set of 180 *L. pneumophila* strains from the West Bank was described previously (16). A comparison with the international 16S rRNA and the sequenced *L. pneumophila* from DUWL in the West Bank showed that *L. pneumophila* was clearly identified and distinguished from other species. *L. pneumophila* is forming a distinct clade, allowing the design of species-specific primers and probes [2,45,49,50]. This demonestrates the higher discriminatory power and effective resolution of using *L. pneumophila*-specific primer in this study. 

## 5. Conclusions

A study on *L. pneumophila* populations in DUWL and dental unit biofilms in the water systems of five dental clinics and faculties across the West Bank demonstrated a moderate abundance of culturable *L. pneumophila* in water and in biofilm. PCR-based analysis consistently showed a higher detection rate in water and biofilm. 16S rRNA sequencing of *L. pneumophila* strains provides an adequate resolution and thus, a good basis for detailed studies of the health- and water-management-relevant traits of *L. pneumophila* in support of a better clinical and DUWL management in the West Bank [51]. Also, incorporating molecular methods into a water safety planning approach for *Legionella* management in dental clinics can greatly enhance the ability to detect, monitor, and control the presence of *Legionella* in water systems. Here is how molecular methods can be integrated into a comprehensive water safety plan for *Legionella* management in dental clinics: (i) Conduct a thorough risk assessment of the dental clinic’s water system to identify potential sources of *Legionella* contamination. This should include an evaluation of the water distribution system, sources of incoming water, and potential areas of stagnation. (ii) Implement a routine water sampling and monitoring program using molecular methods. This involves collecting water samples from critical points in the dental clinic’s water system and analyzing them for the presence of *Legionella* DNA. (iii) Ensure that dental clinic staff are educated and trained in water safety procedures, including the importance of *Legionella* monitoring and the proper response to elevated levels. Incorporating molecular methods into a water safety planning approach for *Legionella* management in dental clinics provides a more sensitive and accurate means of detecting the presence of *Legionella*. This proactive approach helps reduce the risk of LD outbreaks associated with dental clinic water systems and ensures the safety of both patients and staff.

## Figures and Tables

**Figure 1 tropicalmed-08-00490-f001:**
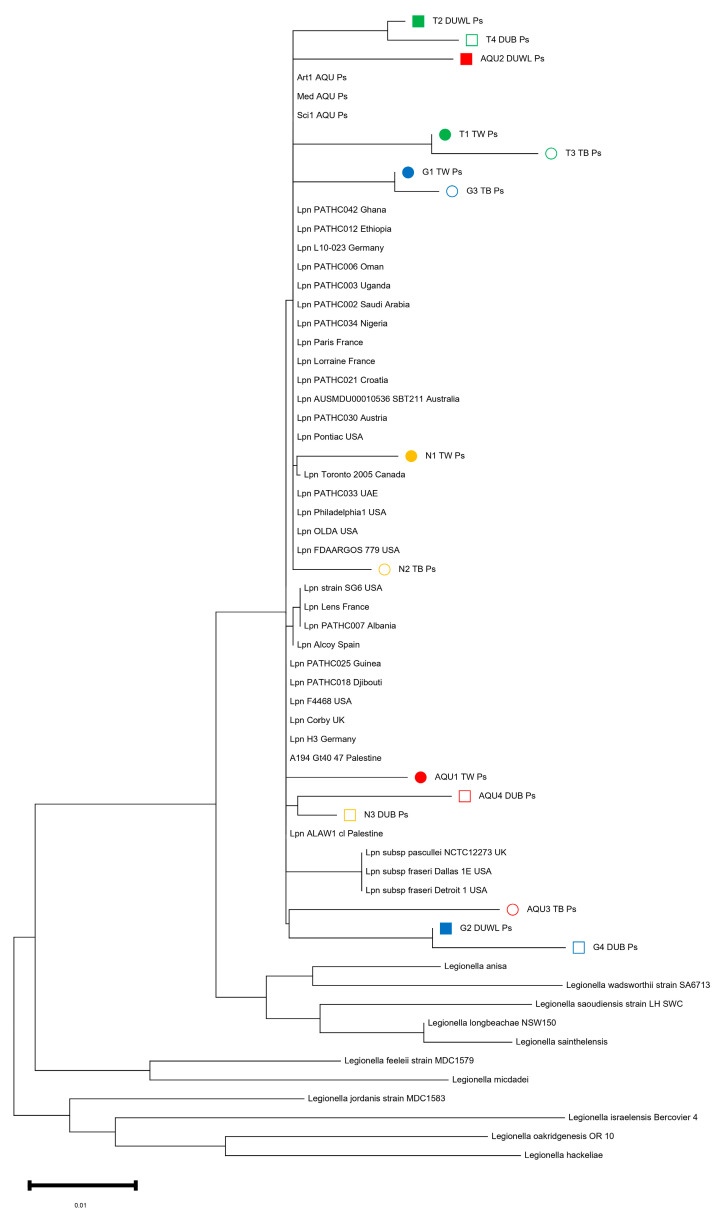
A phylogenetic tree produced with the Maximum Likelihood (ML) method based on the 16S rRNA gene sequences of *Legionella* sp. from dental water and Biofilm samples.

**Table 1 tropicalmed-08-00490-t001:** Average of *Legionella* counts and physico-chemical parameters from the water systems of five dental areas in the West Bank.

Sampling Site (North to South)	Coordinates	Water Type	*Leg*. Count (CFU/L) ± SD	Temperature °C ± SD	pH ± SD	Conductivity µS/cm ± SD	Chlorine mg/L ± SD	Hardness mg/L ± SD
Jenin (Dental Faculty)	32°23′ N, 35°19′ E	Tap Water	57 ± 12	18 ± 0.7	7.8 ± 0.2	856 ± 51	0.5 ± 0.3	278 ± 45
		DUWL	53 ± 6	19 ± 0.5	7.8 ± 0.2	862 ± 54	0.5 ± 0.3	264 ± 32
Tulkarem (Dental clinics)	32°31′ N, 32°18′ E	Tap Water	115 ± 11	24 ± 0.7	7.2 ± 0.1	819 ± 28	0.5 ± 0.3	234 ± 31
		DUWL	42 ± 5	24 ± 0.8	7.2 ± 0.1	828 ± 41	0.5 ± 0.3	244 ± 38
Nablus (Dental clinics)	32°22′ N, 32°13′ E	Tap Water	27 ± 9	21 ± 0.4	7.4 ± 0.2	721 ± 51	0.5 ± 0.3	284 ± 19
		DUWL	BD	20 ± 0.4	7.3 ± 0.2	728 ± 48	0.5 ± 0.3	291 ± 12
Abu Deis/East Jerusalem (Dental Faculty)	31°75′ N, 35°25′ E	Tap Water	47 ± 5	18 ± 0.5	7.8 ± 0.2	724 ± 49	0.5 ± 0.1	241 ± 11
		DUWL	54 ± 12	18 ± 0.4	7.7 ± 0.2	728 ± 21	0.5 ± 0.1	235 ± 18
Hebron (Dental clinics)	31°52′ N, 31°31′ E	Tap Water	BD	17 ± 0.2	7.9 ± 0.3	762 ± 12	0.5 ± 0.1	261 ± 62
		DUWL	BD	18 ± 0.2	7.9 ± 0.2	757 ± 33	0.5 ± 0.1	266 ± 27

DUWL: Dental unit water line; BD: Below detection limit (<5 CFU/L).

**Table 2 tropicalmed-08-00490-t002:** Occurrence frequencies of *Legionella* in water and biofilm samples obtained from five dental sites in the West Bank.

CDA ^1^					CIA ^2^			
Sampling Site	*L. pneumophila* Isolates/Total Number ^3^	% of Isolates	*Leg*. Counts (Mean of Tap Water) (CFU/L) ± SD	*Leg*. Counts (Mean of DUWL) (CFU/L) ± SD	*Legionella* spp. (Lgsp) Positive Samples/Total Number	% of Positive Samples	*L. pneumophila* (Lpn) Positive Samples/Total Number	% of Positive Samples
Jenin	11/56	19.6	57 ± 12	53 ± 6	43/56	76.8	43/56	76.8
Tulkarem	7/31	22.6	115 ± 11	42 ± 5	29/31	93.5	26/31	83.9
Nablus	4/46	8.7	27 ± 9	BD	37/46	80.4	36/46	78.3
Abu Deis/East Jerusalem	6/44	13.6	47 ± 5	54 ± 12	34/44	77.3	34/44	77.3
Hebron	0/8	0.0	BD	BD	4/8	50.0	3/8	37.5
Total	28/185	15.1	NA	NA	147/185	79.5	142/185	76.8
Sample Type								
Tap water	7/39	17.9	NA	NA	35/39	89.7 *	32/39	82.1
DU water	6/50	12	NA	NA	27/50	54 *	27/50	54
Total water	13/89	14.6	NA	NA	62/89	69.7	59/89	66.3
Tap biofilm	9/45	20	NA	NA	41/45	91.1	41/45	91.1
DU biofilm	6/51	11.8	NA	NA	44/51	86.3	42/51	82.4
Total biofilm	15/96	15.6	NA	NA	85/96	88.5	83/96	86.5

NA: Not Available; BD: Below detection limit (<5 CFU/L); DU: Dental Unit; ^1^ Cultivation Dependent Analysis. ^2^ Cultivation Independent Analysis; ^3^ The number of isolates corresponds to the number *L. pneumophila* culture-positive water and biofilm samples. The * indicates statistically significant differences *p ≤* 0.01; statistically significant difference (Independent *t*-test) between tap water and DUWL.

**Table 3 tropicalmed-08-00490-t003:** Serogroup of 28 *L. pneumophila* isolates from the West Bank.

Sampling Site	No. of Isolates	No. Sg 1 (%)	No. Sg 2–14 (%)
Jenin	11	9 (81.8%)	2 (18.2%)
Tulkarem	7	6 (85.7%)	1 (14.3%)
Nablus	4	3 (75%)	1 (25%)
Abu Deis/East Jerusalem	6	3 (50%)	3 (50%)
Hebron	0	0	0
Total	28	21 (75%)	7 (25%)

Sg: Serogroup.

## Data Availability

16S rRNA sequence data are available at NCBI GenBank under BioProject accession number PRJNA1002630. A detailed list of all sequences accession numbers is provided in Appendix A.

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
