# Peer review of "Legionella pneumophila Presence in Dental Unit Waterlines: A Cultural and Molecular Investigation in the West Bank, Palestine"

_tropicalmed, 2023, doi:10.3390/tropicalmed8110490_

Round 1

Reviewer 1 Report

Comments and Suggestions for Authors

Review: Cultural and Molecular Evidence of Legionella pneumophila in Dental Unit Waterlines in the West Bank, Palestine

This paper reports the occurrence, abundance and identity of Legionella spp. in water in dental clinics in Palestine. The study employed both cultural and molecular assays and generated original data that can be used to develop monitoring schemes based on either type of assay.

Although the manuscript presents some interesting data, it lacks the context and analysis that would allow the manuscript to meet it potential. Two areas of further analysis could significantly improve the paper. First, it is not surprising that L. pneumophila was found via culture techniques and that it was found more frequently via molecular methods. Papers, including ones focused on Legionella in dental clinics, have already published this finding. To make the paper more useful the authors are encouraged to investigate more deeply the specific features of the dental clinics that make Legionella more or less common or abundant. This will be difficult for the current study because a high proportion of tap water samples have Legionella and L. pneumophila and it is likely that the entire building water systems for the clinics are contaminated. Second, the paper could explore how molecular methods could be incorporated into a water safety planning approach to Legionella management in dental clinics. The presentation of data and conclusions in the paper do not explore this possibility and leave the reader with no reason to use molecular methods.

Generally, the manuscript will require substantial copy editing before publication, including removal of incorrectly capitalized words, completion of incomplete sentences, correction of many misspellings and many other revisions.

Specific comments

Line 38 – Sporadic and outbreak Legionella infections also occur for immunocompetent people, but not as commonly as for immunocompromised people. Please revise to remove the implication that LD only occurs among immunocompetent people.

Lines 40-41 – Grammatical errors

Line 46 (and a general statement) – You state that Legionella grow in dental lines. This is only true if there are no controls in place. Please add “if not controlled” to the end of the sentence. Elsewhere in the document, please note that there are proven strategies for Legionella control in building water systems including water systems in dental clinics.

Line 57 – I recommend replacing the word “estimate” with “monitor and control”

Line 66 – please be specific about the techniques for Legionella pneumophila control in DUWLs. Also be specific on the role of monitoring.

Line 72 – “This approach” does not refer back to any approach. Is it the approach for control? Or for transmission? Please clarify.

Line 99 – In future research, longer and higher velocity clearing of lines for the municipal water samples is required for a true sample of the supply water.

Section 2.2 (line 119) – Please provide details on chlorine measurement. What type of chlorine (monochloramine or free chlorine) was in each supply and what measurement technique was used?

Section 2.3 (line 126) – Please provide more details and a clearer description of how Legionella pneumophila were differentiated from Legionella spp. Not all agglutination kits detect all of the most common species.

Section 2.7 (line 183) – describe how non-detect observations were handled in statistical analyses.

Line 196 – I have sampled many, many water systems and do not believe it is possible for chlorine to be so consistent at 0.5 mg/L across all the dental clinics, sampling events and sample types. The relationship between chlorine and Legionella control is critical and chlorine must be measured carefully and documented more completely. Please review the chlorine data and measurement techniques.

Section 3.1 (line 190)- For all of the clinics with detections, the tap water samples have higher or comparable Legionella counts compared with the DUWLs. This indicates the entire dental clinic water systems are likely contaminated. Please conduct statistical analyses to evaluate whether there is a statistically significant difference in contamination (% positive samples) and concentration of Legionella for tap water and DUWL samples. Also, report whether the clinics have water safety plans, water management programs or other Legionella control strategies in place.

Section 3.416 (line 244). This section was difficult to follow and needs to be rewritten. Also, there does not seem to be any conclusion or useful analysis of the results. Please rewrite the section for clarity and more clearly state the importance of the findings.

Line 295 – I disagree that the occurrence was low. The occurrence is far higher than allowable by any existing protocol for assessing control.

Lines 323 – 334 – This study did not identify DUWLs as a specific problem. Instead, it showed that tap water (the supply of DUWLs) was generally contaminated for the sample sites.

Line 387 – Please describe how patients can be protected and how Legionella can be controlled in buildings and health care facilities. These are both known and are important context for the discussion.

Line 404 – 407 – The study described at the end of the paragraph does not seem to relate to anything in the paragraph.

Comments on the Quality of English Language

The English language requires substantial editing.

Author Response

Review: Cultural and Molecular Evidence of Legionella pneumophila in Dental Unit Waterlines in the West Bank, Palestine

This paper reports the occurrence, abundance and identity of Legionella spp. in water in dental clinics in Palestine. The study employed both cultural and molecular assays and generated original data that can be used to develop monitoring schemes based on either type of assay.

Although the manuscript presents some interesting data, it lacks the context and analysis that would allow the manuscript to meet it potential. Two areas of further analysis could significantly improve the paper. First, it is not surprising that L. pneumophila was found via culture techniques and that it was found more frequently via molecular methods. Papers, including ones focused on Legionella in dental clinics, have already published this finding. To make the paper more useful the authors are encouraged to investigate more deeply the specific features of the dental clinics that make Legionella more or less common or abundant. This will be difficult for the current study because a high proportion of tap water samples have Legionella and L. pneumophila and it is likely that the entire building water systems for the clinics are contaminated. Second, the paper could explore how molecular methods could be incorporated into a water safety planning approach to Legionella management in dental clinics. The presentation of data and conclusions in the paper do not explore this possibility and leave the reader with no reason to use molecular methods. More text added L527-543

Generally, the manuscript will require substantial copy editing before publication, including removal of incorrectly capitalized words, completion of incomplete sentences, correction of many misspellings and many other revisions.

Specific comments

Line 38 – Sporadic and outbreak Legionella infections also occur for immunocompetent people, but not as commonly as for immunocompromised people. Please revise to remove the implication that LD only occurs among immunocompetent people.

Change to occurs via inhalation. L42

Lines 40-41 – Grammatical errors

Done- L43

Line 46 (and a general statement) – You state that Legionella grow in dental lines. This is only true if there are no controls in place. Please add “if not controlled” to the end of the sentence. Elsewhere in the document, please note that there are proven strategies for Legionella control in building water systems including water systems in dental clinics.

Added- if not controlled. L53

Line 57 – I recommend replacing the word “estimate” with “monitor and control”

Done- L62

Line 66 – please be specific about the techniques for Legionella pneumophila control in DUWLs. Also be specific on the role of monitoring.

Done- L72-83

Line 72 – “This approach” does not refer back to any approach. Is it the approach for control? Or for transmission? Please clarify.

Done L90

Line 99 – In future research, longer and higher velocity clearing of lines for the municipal water samples is required for a true sample of the supply water.

Ok, Thank you

Section 2.2 (line 119) – Please provide details on chlorine measurement. What type of chlorine (monochloramine or free chlorine) was in each supply and what measurement technique was used?

Done. L145-150

Section 2.3 (line 126) – Please provide more details and a clearer description of how Legionella pneumophila were differentiated from Legionella spp. Not all agglutination kits detect all of the most common species.

I think you mean section 2.5- I added more text L197-204

Regarding differentiation between L. pneumophila and Legionella spp this performed as the following:

  • By CIA: using specific primers
  • Legionella isolates differentiated by Oxoid latex agglutination

Regarding section 2.3 CDA: In this section we isolated Legionella spp from different sample types (water/biofilm) then these isolates were identified by latex agglutination and 16S rRNA sequencing.

Section 2.7 (line 183) – describe how non-detect observations were handled in statistical analyses.

Done. L222-223

Line 196 – I have sampled many, many water systems and do not believe it is possible for chlorine to be so consistent at 0.5 mg/L across all the dental clinics, sampling events and sample types. The relationship between chlorine and Legionella control is critical and chlorine must be measured carefully and documented more completely. Please review the chlorine data and measurement techniques.

I totally agree with you. L231 – constant was deleted

In this manuscript and also in a previous publication (10.3390/pathogens9121012) supplementary material. I didn’t find a clear difference between chlorine levels. This is might be due to the measuring method which is semi-quantitative strips method depends on changing of color and determined by eye.

At the moment, I am working on environmental samples and I measuring both free chorine and chloramine using spectrophotometer colorimetric method. I can say, yes there is a difference.

Unfortunately, I this manuscript I don’t have such a results

Section 3.1 (line 190)- For all of the clinics with detections, the tap water samples have higher or comparable Legionella counts compared with the DUWLs. This indicates the entire dental clinic water systems are likely contaminated. Please conduct statistical analyses to evaluate whether there is a statistically significant difference in contamination (% positive samples) and concentration of Legionella for tap water and DUWL samples. Also, report whether the clinics have water safety plans, water management programs or other Legionella control strategies in place.

Done. Table 2 and  

Also, I sun other statistical analysis tests (Ex: Regression) unfortunately, I didn’t find a clear correlation in contamination and Legionella counts for tap water and DUWL.

L249-252

Section 3.416 (line 244). This section was difficult to follow and needs to be rewritten. Also, there does not seem to be any conclusion or useful analysis of the results. Please rewrite the section for clarity and more clearly state the importance of the findings.

Done. The whole section reworked L297-341

Line 295 – I disagree that the occurrence was low. The occurrence is far higher than allowable by any existing protocol for assessing control.

Low is deleted. L371

Lines 323 – 334 – This study did not identify DUWLs as a specific problem. Instead, it showed that tap water (the supply of DUWLs) was generally contaminated for the sample sites.

Done. L425-427

Line 387 – Please describe how patients can be protected and how Legionella can be controlled in buildings and health care facilities. These are both known and are important context for the discussion.

Done.L480-497

Line 404 – 407 – The study described at the end of the paragraph does not seem to relate to anything in the paragraph.

 Deleted L514-518

Reviewer 2 Report

Comments and Suggestions for Authors

to authors;

The study is interesting because it successfully explains a problem that dentists face professionally. At the same time, the study presents local data.

Numerous spelling errors in the text should be corrected. eg.  L. Pneumophila -> L. pneumophila ; analyses -> analysis ; previious -> previous etc

The text "More details on cultivation dependent analyzes are given by" in line 139 should be changed to ".... by Zayed et. al (16)". Additionally, "analyses" should be "analysis".

Please add your comments to the discussion as to why your CDA isolates were found to be low.

Please when writing the sources, ensure that you write them according to the rules. All sources should be written in the same template.

---

Comments on the Quality of English Language

Minor editing of English language required eg. spelling and text fluency

Author Response

The study is interesting because it successfully explains a problem that dentists face professionally. At the same time, the study presents local data.

Numerous spelling errors in the text should be corrected. eg.  L. Pneumophila -> L. pneumophila ; analyses -> analysis ; previious -> previous etc

Corrected

The text "More details on cultivation dependent analyzes are given by" in line 139 should be changed to ".... by Zayed et. al (16)". Additionally, "analyses" should be "analysis".

Done

Please add your comments to the discussion as to why your CDA isolates were found to be low.

Done. L399-414

Please when writing the sources, ensure that you write them according to the rules. All sources should be written in the same template.

---

Comments on the Quality of English Language

Minor editing of English language required eg. spelling and text fluency

Submission Date

05 September 2023

Date of this review

12 Sep 2023 19:33:22

Reviewer 3 Report

Comments and Suggestions for Authors

In manuscript "Cultural and Molecular Evidence of Legionella pneumophila in 2 Dental Unit Waterlines in the West Bank, Palestine"the authors show the presence of legionella in dental unit waterlines in Palestine. It is already a well-known fact that Legionella are found in artificial water systems and that they are very often found in dental unit waterlines, but there may be differences in different parts of the world. For this reason, I believe that the study of the prevalence of legionella is important. Materials and methods are well and thoroughly described. The only thing I would comment on is that swabbing as a method of biofilm sampling has its drawbacks and that scraping the surface would be a better option and that even higher results would be obtained. Please consider it in the discussion.

I have no comments on the presentation of the results and their discussion.

Author Response

In manuscript "Cultural and Molecular Evidence of Legionella pneumophila in 2 Dental Unit Waterlines in the West Bank, Palestine"the authors show the presence of legionella in dental unit waterlines in Palestine. It is already a well-known fact that Legionella are found in artificial water systems and that they are very often found in dental unit waterlines, but there may be differences in different parts of the world. For this reason, I believe that the study of the prevalence of legionella is important. Materials and methods are well and thoroughly described. The only thing I would comment on is that swabbing as a method of biofilm sampling has its drawbacks and that scraping the surface would be a better option and that even higher results would be obtained. Please consider it in the discussion.

Added L398-400

I have no comments on the presentation of the results and their discussion.

Submission Date

05 September 2023

Date of this review

06 Oct 2023 10:44:17

Round 2

Reviewer 1 Report

Comments and Suggestions for Authors

The authors have addressed my comments adequately and made this paper more impactful. 

Comments on the Quality of English Language

The paper should be edited thoroughly for grammar and typographic errors.